# Assessment of urban aerosol pollution over Moscow megacity by MAIAC aerosol product.

Ekaterina Yu. Zhdanova[1], Natalia Ye. Chubarova[1], Alexei I. Lyapustin[2]

[1]Department of Meteorology and Climatology, Faculty of Geography, Lomonosov Moscow State University, Moscow, 119991, Russia
[2]Laboratory for Atmospheres, NASA Goddard Space Flight Center, Greenbelt, Maryland, USA

*Correspondence to*: Ekaterina Yu. Zhdanova (ekaterinazhdanova214@gmail.com)

**Abstract.** We estimated the distribution of aerosol optical thickness (AOT) with a spatial resolution of 1 km over Moscow megacity using MAIAC aerosol product based on MODIS satellite data (Lyapustin et al., 2018) for the warm period of year (May-September, 2001-2017). AERONET (Aerosol Robotic Network)-based validation of satellite estimates near the city centre at Moscow_MSU_MO and over Moscow suburbs at Zvenigorod revealed that MAIAC AOT at 0.47 μm is in agreement with AERONET AOT though underestimated by 0.05-0.1 for AOT<1 and overestimated for smoke conditions with AOT>1. The MAIAC AOT biases were almost the same for the Moscow_MSU_MO and Zvenigorod AERONET sites, which indicated that MAIAC algorithm effectively removed the effect of the bright urban surface in the city centre. For the ground-based measurements, the difference between annual median AOT at Moscow_MO_MSU and Zvenigorod (ΔAOT) varied within -0.002-+0.03 with statistically significant positive bias for most years, and an average ΔAOT was about 0.02. According to the MAIAC dataset, the ΔAOT varied within ±0.01 and were not statistically significant. The ΔAOT started decreasing recently due to intensive urban development of the territory around Zvenigorod and the decrease of pollutant emissions in Moscow, which is mainly caused by the environmental regulations. According to the MAIAC dataset, the most pronounced spatial AOT differences over the territory of Moscow was observed at 5% quantile level, where they reached 0.05-0.06 over several locations and could be attributed to the stationary sources of aerosol pollution, for example, large areas of construction sites, aerosol pollution from roads and highways, or agriculture activities. The differences between the maximum and the mean AOT for different quantiles, except the 95% quantile, within the Moscow region, were about 0.02-0.04 which could be attributed to the local aerosol sources. The application of the MAIAC algorithm over the whole Moscow region has revealed a decreasing AOT trend over the centre of Moscow and an increasing trend over the "New" Moscow territory which experienced an intensive build-up and agricultural development.

## 1 Introduction

Atmospheric aerosols are the suspended particulate components of the atmosphere, which are produced directly from the emissions of particulate matter of different origins and generated from gaseous precursors. The variety of chemical and physical processes of aerosol formation provides a large diversity of their microphysical and optical properties. A significant

variation of aerosol properties has been observed in the industrial urban areas. Anthropogenic aerosols affect the temperature profile, play important role as a cloud condensation nuclei, impact the hydrologic cycle, through changes in cloud cover, cloud properties and precipitation (Kaufman et al., 2002, Kaufman, 2006).

One of the key aerosol optical characteristics is the aerosol optical thickness (AOT), whose spatial and temporal variations have been studied using satellite and ground-based data in numerous papers (Koelemeijer et al., 2006, Schaap et al., 2008, Chubarova, 2009, Bovchaliuk et al., 2013, Putaud et al., 2014, Chubarova et al., 2016, etc.). Over the Europe, a permanently elevated aerosol loading was observed over several industrial regions with particularly high values found over Netherlands, Belgium, the Ruhr area, the Po-valley, the Northern Germany and the former East Germany, Poland, and parts of Central European countries. Elevated aerosol loading usually correlates with a suspended particulate matter associated with the poor air quality (Wang and Christopher, 2003, Hoff, Christopher, 2009, Chudnovsky et al., 2012, van Donkelaar et al., 2015). Recently a high 1 km resolution aerosol MAIAC satellite product has been used for estimating relationships between AOT and particulate matter (Chudnovsky et al., 2013b, Hu et al., 2014, Kloog et al., 2015, Xiao et al., 2017, Beloconi et al., 2018, Liang et al., 2018, Han et al., 2018).

Large cities with their high road density and industrial enterprises are the source of aerosol pollution, which includes black carbon, sulphate, nitrate and ammonium aerosol components as well as primary and secondary organic aerosols (POA and SOA) (IPCC, 2013). And the urban aerosol is dominated by the fine mode particles (Kaufman et al., 2005).

Several recent studies reported an analysis of AOT based on ground-based and satellite data over Moscow (Chubarova et al. 2011a, Kislov, 2017), Warsaw (Zavadzka et al, 2013), Córdoba (central Argentina) (Della Ceca et., 2018) urban areas.

Previously, the urban aerosol pollution in Moscow has been studied using concurrent observations by the AERONET Cimel sun- photometers located in the Moscow city and in the suburbs (Zvenigorod). This study revealed an average AOT at 0.5 μm of ~0.19 of which 0.02 was apportioned to the urban sources, and a tendency of lower single scattering albedo (higher absorption) in Moscow (Chubarova et al., 2011a). The difference between AOT in the city of Warsaw and suburban conditions of Belsk was estimated as 0.02 (at 0.5 μm) based on sun photometers' data (Zawadzka et al., 2013). However, the use of only two contrasting ground-based sites does not allow assessing the detailed spatial distribution of AOT and estimating an integrated urban aerosol loading even at high quality of the AOT measurements. This task can be solved by using high quality satellite AOT retrievals.

The analysis of the results obtained from the Visible Infrared Imaging Spectrometer (VIIRS) (Jackson et al., 2013) showed that the central part of the Moscow city has a significantly higher AOT at 0.55 μm (by about 0.1) than that in the suburbs (Zhdanova, Chubarova, 2018). Such a significant difference, as discussed in this paper, has probably originated from the uncertainty in evaluation of the urban surface reflectance in the VIIRS aerosol algorithm (Liu et al., 2014). The assessment of the aerosol pollution in Moscow using the mid-visible range AOT from the MODIS data (collection 5.1) with a 1º ×1º spatial resolution during the warm period of 2000-2013 showed that the difference in AOT due to urban effects can reach up to 0.08 if compared to AOT obtained over the green areas to the north of 58º N or to the south of 53º N (Kislov, 2017). However, the spatial resolution and the uncertainties of the AOT retrievals used in this study did not allow determining the

detailed spatial features of AOT distribution. The MAIAC aerosol product (Lyapustin et al., 2018), based on MODIS data, has some advantages over the standard MODIS algorithms: it overcomes empirical assumptions related to surface reflectance and provides AOT at high 1 km spatial resolution. MAIAC uses the minimum reflectance method, implemented dynamically, to separate atmospheric and surface contributions. The sliding window technique, accumulating a time series of data for up to 16-days, provides a necessary surface characterization via dynamic retrieval of the spectral bidirectional

reflectance distribution function (BRDF) (Lyapustin et al., 2018). A good knowledge of surface BRDF allows MAIAC to minimize effects of both surface brightness and view geometry on MAIAC AOT as compared to the standard MODIS Dark Target (DT) and Deep Blue (DB) products (e.g., Mhawish et al., 2018; Jethva et al., 2019).

    Thus, the objective of this paper is to verify the MAIAC aerosol retrievals against the ground-based AERONET measurements over the Moscow area (for the urban and suburban sites) and to evaluate the temporal trends and spatial

features of the urban aerosol pollution over the Moscow megacity for the time period from 2001 to 2017.

## 2. The study area, datasets and methodology

### 2.1 The study area

The Moscow megacity (55º45′N, 37º 37′E) is one of the largest urban agglomerations in the world with population of 12.6 million according to the Federal Statistics Service (on January 1st, 2019) with industrial enterprises and technologies in the

field of mechanical engineering and metalworking, energy and petrol chemistry, light and food industries, construction materials and an intensive residential development (Kulbachevski, 2018). In 2012, the Moscow megacity has expanded mostly to the south-west to include a "New" Moscow region. As a result, its territory has increased from 1091 to 2511 km$^2$ (https://www.mos.ru/en/). The study domain is shown in Fig. 1. The Moscow city boundaries, its administrative districts and satellite image of Moscow region are shown in Fig.1.

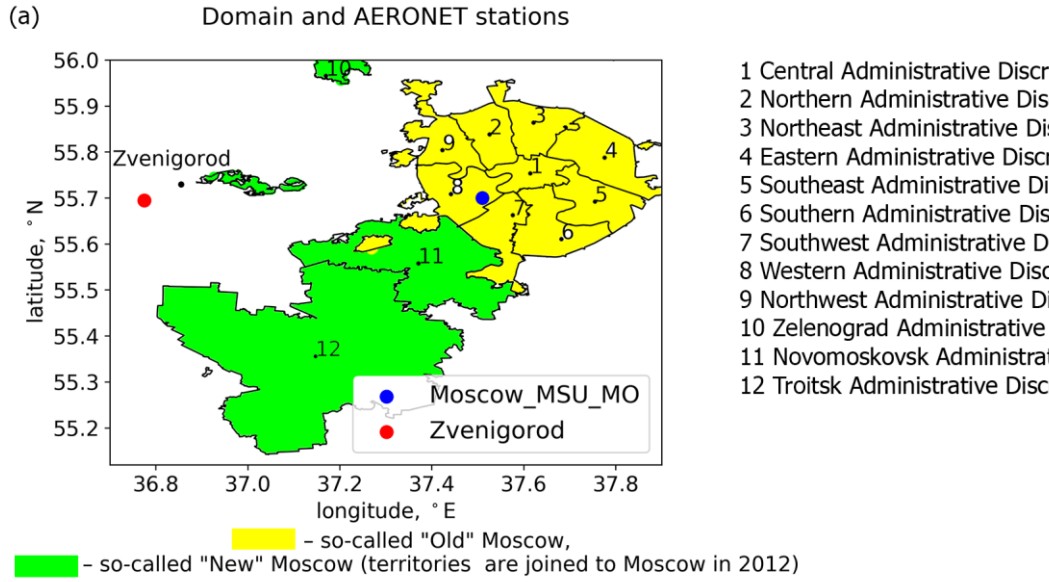

(a) Domain and AERONET stations

1 Central Administrative Discrict
2 Northern Administrative Discrict
3 Northeast Administrative Discrict
4 Eastern Administrative Discrict
5 Southeast Administrative Discrict
6 Southern Administrative Discrict
7 Southwest Administrative Discrict
8 Western Administrative Discrict
9 Northwest Administrative Discrict
10 Zelenograd Administrative Discrict
11 Novomoskovsk Administrative Discrict
12 Troitsk Administrative Discrict

⬜ – so-called "Old" Moscow,
🟩 – so-called "New" Moscow (territories are joined to Moscow in 2012)

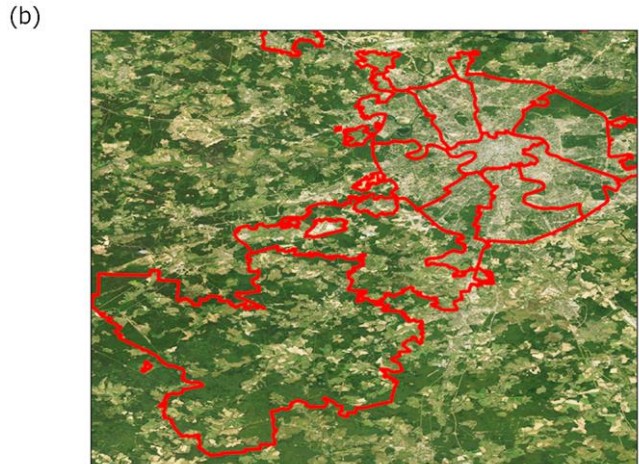

(b)

**Figure 1. Study domain and location of AERONET sites.**

    a) **"Old" and "New" Moscow, administrative districts**
    b) **Satellite image (ArcGIS World Imagery - https://arcg.is/4zubf)**

## 2.2. MAIAC data

100    A new MODIS satellite product - MCD19A2 Collection 6 (MAIAC aerosol product) with 1 km spatial resolution was used to estimate spatial-temporal distribution of AOT over the Moscow region (https://search.earthdata.nasa.gov/search).

MCD19A2 product provides a suite of atmospheric parameters and view geometry, including: column water vapor, AOT at 0.47 and 0.55 μm, AOT uncertainty, fine mode fraction over water, smoke injection height (m above ground), AOT QA (Quality Assurance), AOT model at 1km, and a view geometry suite at 5 km (cosine of solar zenith angle, cosine of view zenith angle, relative azimuth angle, scattering angle, and glint angle). Each parameter within each MCD19A2 Hierarchical Data Format 4 (HDF4) file contains a third dimension that represents the number of orbit overpasses. We used the data for the warm snow-free time period from May to September over the 2001-2017 years. The geographical location of the Moscow region corresponds to the MODIS granule h20v03. The MAIAC algorithm retrieves AOT at 0.47 μm and provides an additional value at the standard wavelength 0.55 μm calculated according to the aerosol model used. MAIAC uses 8 different regional background aerosol models tuned to the AERONET (Aerosol Robotic Network, (Holben et al., 1998)) climatology. Each geographical location has one predefined aerosol model. Aerosol model number 1 is used for Moscow region. The MAIAC algorithm also detects absorbing dust and smoke aerosols and provides dust/smoke mask in the QA. The smoke test relies on a relative increase in aerosol absorption at MODIS wavelength 412 nm compared to 470–670 nm owing to multiple scattering and enhanced absorption by organic carbon released during biomass burning combustion (Lyapustin et al., 2012). A detailed description of the MAIAC aerosol algorithm can be found in (Lyapustin et al., 2018). Only AOT values with the highest quality were used in the presented analysis (QA.QA_AOT = Best_Quality).

## 2.3 AERONET data

The data from the two sites equipped with the Cimel sun/sky photometers of the AERONET project (Holben et al., 1998) were used for validation of the satellite AOT retrievals, as well as for determining the features of the AOT temporal-spatial distribution over the territory of Moscow megacity. They included the measurements of the Observatory of Moscow State University (Moscow_MSU_MO site, 55.70695° N, 37.52202° E) over the 2002-2017 period and Zvenigorod scientific station of Institute of Atmospheric Physics, Russian Academy of Sciences (Zvenigorod site, 55.695° N, 36.775° E) over the 2006-2017 period. The first site is located within the city, at a distance of about 8 km from the city centre, the second - the upwind suburban area about 50 km west from the city centre. The AERONET measurements at level 2, version 3 (Giles et al., 2019) were used with the additional cloud-screening using ground-based visual cloud observations at the Meteorological Observatory of Moscow State University, as described in (Chubarova et al., 2016). Long-term measurements at the Moscow_MSU_MO have revealed noticeable seasonal changes in AOT with maximum in April and July with median AOT at 0.5 μm of about 0.22, and minimum in December and January with AOT at 0.5 μm of 0.07 (Chubarova et al. 2011b, Chubarova et al., 2016). However, in this study we focused on snow-free period (May-September), during this period of year AOT variations are not large (~ 0.15-0.21). Additionally, we used AERONET estimates of fine mode fraction (O'Neill et al., 2003). The location of the AERONET sites are shown in Fig.1.

## 2.4 EMEP data

In addition, we used the EMEP ('European Monitoring and Evaluation Programme') grid archive (http://www.ceip.at/new_emep-grid/01_grid_data) for assessing the spatial-temporal distribution of aerosol precursor gases

emissions to explain the spatial features of the AOT distribution. We analysed the main precursor gases $NO_x$, $SO_x$, NMVOC, $NH_3$, along with particulate matter concentrations (PM2.5 and PM10).

## 3. Results

### 3.1 Validation of satellite AOT retrievals against ground-based data.

The MAIAC aerosol algorithm was successfully validated over various geographic regions: over bright desert surfaces
(Sever et al., 2017), over South Asia (India) (Mhawish et al., 2019), over mountainous areas (Emili et al., 2011), across South America (Martins et al., 2017), and over North America (Jethva et al., 2019). Mhawish et al., (2019) gave a detailed comparison of MAIAC data with standard MODIS algorithms and ground-based data, and studied the accuracy of product as a function the sensor (MODIS on Terra or Aqua), the underlying surface, aerosol model, and scanning geometry. According to (Mhawish et al., 2019), the MAIAC AOT error is about 15%. At high AOT, MAIAC underestimates AOT, especially in
MODIS Aqua record (Mhawish et al., 2019). However, on average, the AOT MAIAC data are characterized by smaller errors compared to the two operational MODIS algorithms: Dark Target (Levy et al., 2013) and Deep Blue (Hsu et al., 2013).

We averaged AERONET data to 1-hour resolution and calculated AOT at 0.47 μm from available AERONET AOT at 0.44 μm and Angstrom exponent (0.44-0.87 μm) in this study. MAIAC AOT data were spatially averaged with a 5-km radius
circle centred at the Moscow_MSU_MO and Zvenigorod sites and also averaged within 1 hour to have robust estimates. Correlations are plotted separately for the Terra and Aqua datasets, and together for the data from the two satellites in the 1 hour intervals (Fig. 2). As can be seen in Fig. 2, the satellite AOT at 0.47 μm retrievals for Moscow_MO_MSU and Zvenigorod are underestimated by about -0.05 for the values less than 1, and overestimated in conditions of high aerosol loading in Moscow. However, the correlation between the AOT MAIAC retrievals and AERONET data is high. Slopes of
regressions lines are higher at the Moscow_MO_MSU site than that at Zvenigorod, since at Zvenigorod site high aerosol loading due to forest and peatbog fires has not been included in the sample.

The overestimation of the AOT MAIAC occurs in cases of forest fires, when the MAIAC algorithm detects smoke. This is clearly seen in Fig.3, where the cases of detected smoke are shown by an orange color. Overall, this error is in contrast to the typical biomass burning conditions when the MAIAC algorithm usually underestimates AOT (e.g., see Lyapustin et al.,
2018). The underestimation is caused by the fact that MAIAC C6 algorithm keeps using the same background model in cases of detected smoke which usually has higher absorption for fresh smoke aerosol (Dubovik et al., 2002). On the contrary, the Moscow smoke of 2010 was largely a result of smoldering peat fires producing larger particle size and lower absorption (Chubarova et al., 2012, Sayer et al., 2014), the combination for which led to the AOT overestimation.

Statistical estimates (RMSE - root mean square error, MAE - mean absolute error, BIAS - mean error) of the quality of the
AOT at 0.47 μm retrievals relative to the ground-based AERONET data are presented in Table 1. It is worth noting that the errors of the MAIAC AOT are similar to both Moscow_MSU_MO and Zvenigorod sites which indicates that the bias is

model-related while the contribution of bright urban underlying surface is effectively taken into account in the MAIAC algorithm.

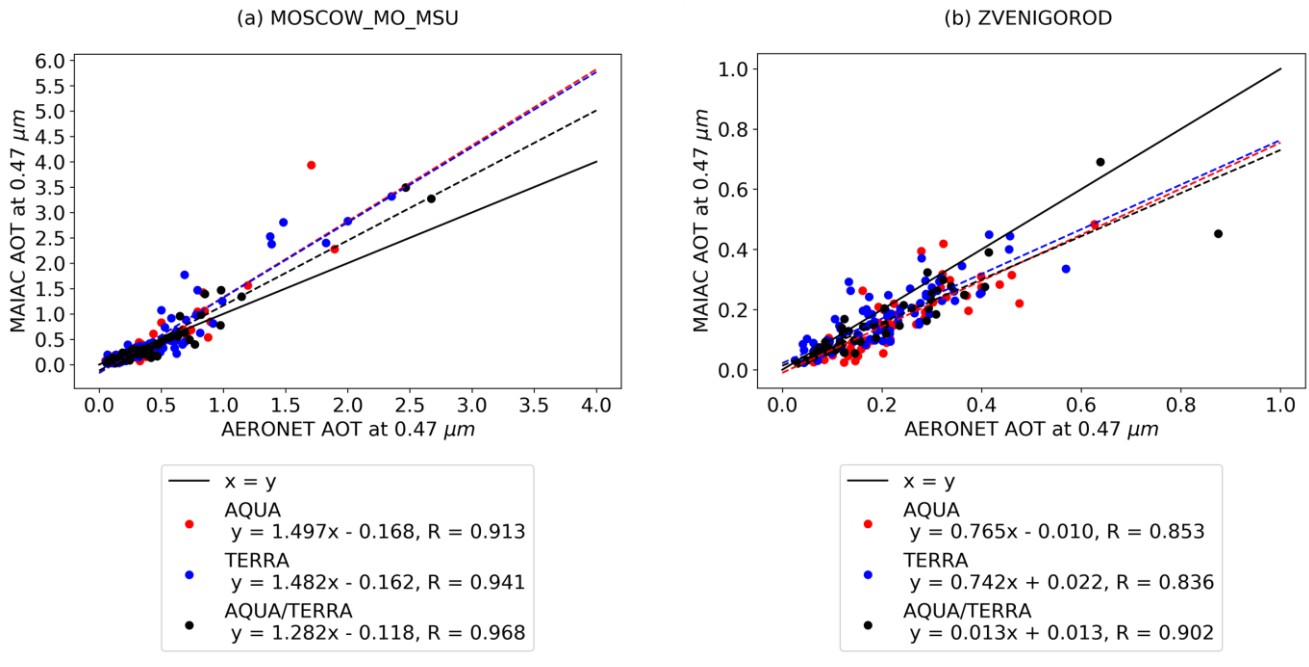


**Figure 2. Correlations between MAIAC AOT at 0.47 μm and AERONET AOT at 0.47 μm for Moscow_MSU_MO and Zvenigorod AERONET sites for Terra, Aqua and their joint overpasses within 1 hour (Aqua/Terra).**
**Comment: the absence of high AOT values at Zvenigorod site is explained by technical problems with the instrument and the absence of the AERONET data at level 2 version 3 in 2010, when intensive forest fires took place.**


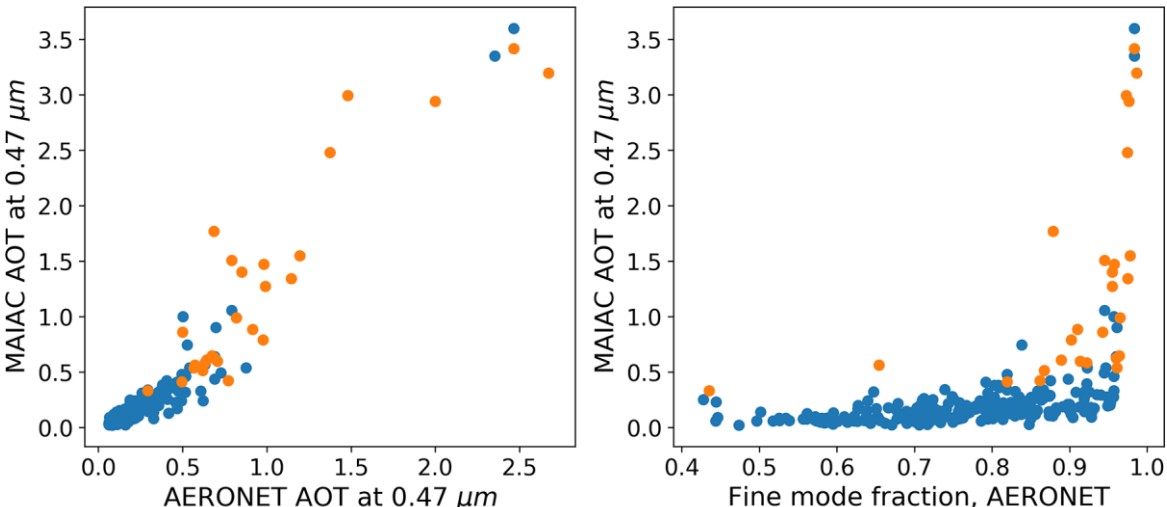

**Figure 3. MAIAC AOT at 0.47 μm against AERONET AOT (left) and MAIAC AOT at 0.47 μm against fine mode fraction AOT AERONET (right) according to the regional MAIAC aerosol model (blue color) and in cases of smoke detection (orange color). Moscow, 2001-2017.**


**Table 1. Statistical estimates of the uncertainties in AOT MAIAC retrievals for the Terra and Aqua data separately, for the Terra and Aqua measurements within 1 hour (Aqua/Terra), and together for the data from the two satellites (Terra and Aqua) against ground-based AERONET data at the MOSCOW_MO_MSU (2001-2017) and ZVENIGOROD (2006-2017) sites.**

**RMSE - root mean square error, MAE - mean absolute error, BIAS - mean error, N - case number.**

| | MOSCOW _MSU_ MO, all AOT | | | |
|---|---|---|---|---|
| | TERRA | AQUA | AQUA/ TERRA | TERRA and AQUA |
| RMSE | 0.24 | 0.23 | 0.17 | 0.22 |
| MAE | 0.12 | 0.1 | 0.09 | 0.11 |
| BIAS | 0 | -0.02 | -0.02 | -0.02 |
| N | 181 | 130 | 99 | 410 |
| | MOSCOW _MSU_ MO, AOT<1 | | | |
| RMSE | 0.1 | 0.09 | 0.1 | 0.1 |
| MAE | 0.07 | 0.07 | 0.07 | 0.07 |
| BIAS | -0.05 | -0.05 | -0.05 | -0.06 |
| N | 171 | 124 | 94 | 389 |
| | ZVENIGOROD, AOT<1 | | | |
| RMSE | 0.07 | 0.09 | 0.08 | 0.08 |
| MAE | 0.05 | 0.07 | 0.05 | 0.06 |
| BIAS | -0.03 | -0.06 | -0.04 | -0.04 |
| N | 77 | 61 | 48 | 186 |

### 3.2 Temporal AOT changes in Moscow according to ground-based and satellite data

We studied temporal AOT changes using MAIAC AOT retrievals and AERONET long-term measurements collocated in time over Moscow_MSU_MO site during a warm May-September period. Fig. 4a shows the time series of AOT at 0.55 μm built for all available Moscow_MSU_MO AERONET and MAIAC data. One can see a satisfactory agreement between the satellite and ground-based observations with the exception of 2002 and 2010 years. The highest AOT were observed in 2010 and 2002 years due to the effects of smoke aerosols from peat and forest fires in Moscow region (Chubarova et al, 2011b). In 2016 the smoke aerosol advection was also observed from the Siberia area (Sitnov et al., 2017) providing an intermediate AOT maximum. Fig.4b shows year-to-year variability of AOT at 0.55 μm only for matching within 1 hour Moscow_MSU_MO AERONET and MAIAC data, and for the cases, when MAIAC regional background aerosol model has been applied. One can see a better agreement between MAIAC AOT and corresponding AERONET AOT data in year-to-year variations. There is a clearly seen decrease in AOT during the last years according to both the MAIAC (when regional model was used) and the AERONET data. The yearly means difference between AERONET and MAIAC data (AOT MAIAC – AOT AERONET) is -0.03 for the all matching data (blue and red lines in Fig 4b) and -0.05 for the matching data with MAIAC regional aerosol model estimates (blue and orange lines in Fig 4b). Fig.4c presents the AOT variations only for the cases of the MAIAC smoke detection. It is seen that the AOT MAIAC overestimation is taken place only for the cases with high AOT>1.

Thus, MAIAC AOT algorithm reproduces the absolute AOT values and the long-term AOT decrease in Moscow for the regional background aerosol model while in case of smoke aerosol detection there is a significant overestimation of the annual AOT mean. Therefore, for the further analysis of urban aerosol pollution, we used only the AOT MAIAC retrievals with its attribution to the regional background model for removing large smoke aerosol effects, which are also characterized by significant spatial inhomogeneity.

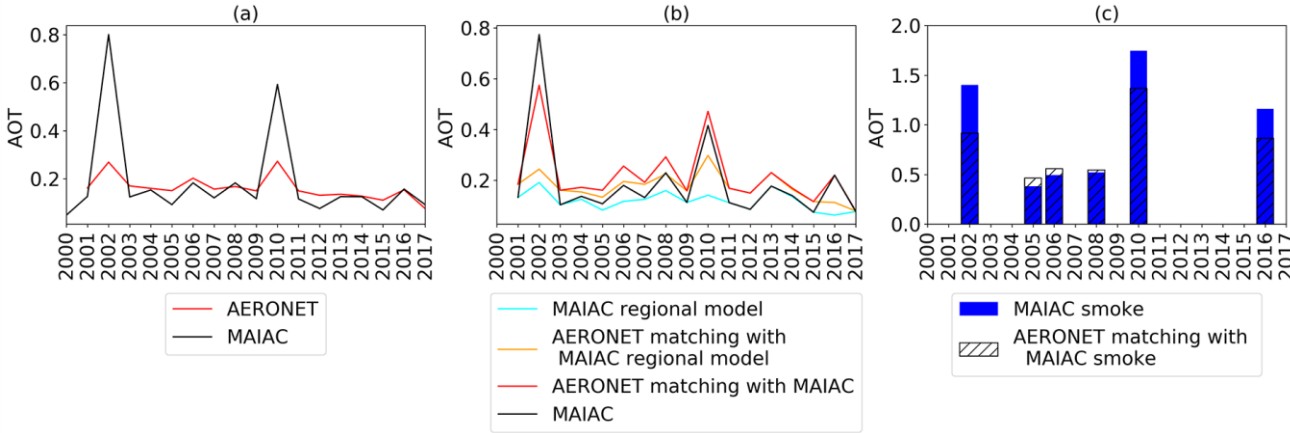

**Figure 4. The year-to-year variations of AOT at 0.55 μm (May-September, mean values) according to AERONET (Moscow_MSU_MO) and MAIAC data: a) all available AERONET and MAIAC data, b) matching AERONET and**

MAIAC data for all cases and for regional aerosol model only, c) AOT MAIAC in cases of smoke detection and matching AERONET data.

**3.3 AOT urban effect according to ground-based and satellite measurements over Moscow_MSU_MO and**
**Zvenigorod AERONET sites.**

Let us consider, how accurately MAIAC can reproduce the urban aerosol effect, which we evaluate as the difference of AOT between Moscow_MSU_MO and Zvenigorod ($\Delta$AOT = AOT (MOSCOW_MO_MSU) – AOT (ZVENIGOROD)). It should be noted that two sites are close enough to each other, so they are influenced by the medium- and long-range transport similarly. Note, that Zvenigorod site has an upwind location. Fig.5 shows a relationship between dAOT from MAIAC and
from hourly-averaged AERONET data. The $\Delta$AOT values obtained from both ground-based and satellite data lie within the range of -0.1 ... 0.1. It should be noted that the $\Delta$AOT between Moscow_MSU_MO and Zvenigorod based on satellite and ground-based data generally correspond to each other. The $\Delta$AOT between the city and the suburbs can be both positive and negative: $\Delta$AOT varies from -0.4 to 0.21 according to ground-based data and from -0.22 to 0.1 according to satellite data (see Fig.5b).


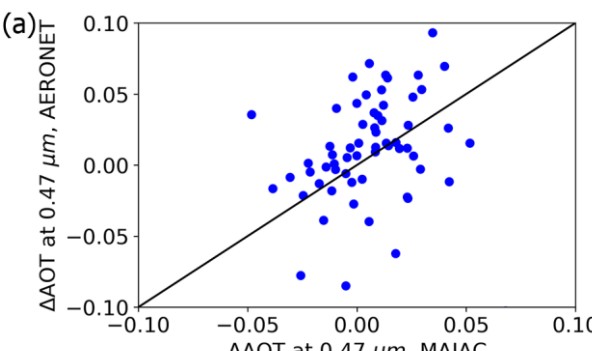
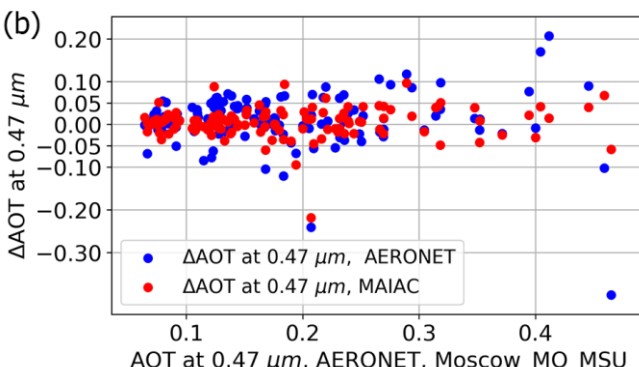

**Figure 5. (a) Relationship between $\Delta$AOT at 0.47µm ($\Delta$AOT=AOT$_{Moscow\_MO\_MSU}$ - AOT$_{Zvenigorod}$) obtained from the satellite and ground-based data; (b) $\Delta$AOT at 0.47µm as a function of AOT at 0.47µm obtained from Moscow_MSU_MO dataset.**


For characterizing variations in $\Delta$AOT we analysed frequency distributions according to ground-based and satellite data. In general, polar orbiting satellites demonstrate similar daily average AOT independent of morning or afternoon orbits (Kaufman et al., 2000). However, we calculated $\Delta$AOT separately for Terra and Aqua datasets for evaluating possible diurnal (in the morning and noon hours) variability of $\Delta$AOT. Frequency distributions of $\Delta$AOT at 0.47 and 0.55 µm
separately for the Terra and Aqua data, and together for the data from the two satellites are shown in Fig.6. The highest

repeatability of ΔAOT is in the range of 0-0.05. For the Aqua AOT retrievals, which are closer to noon, the predominance of positive ΔAOT is more pronounced. Fig. 6 also shows large negative ΔAOT in cases of Terra measurements in our sample. In overall, the ΔAOT at 0.47 values lie within the [0, 0.05] bin in 57% of cases for the Aqua and in 50% - for the Terra datasets.

The diurnal variations of the ΔAOT according to satellite and ground-based data are also shown in Fig.7. The MAIAC ΔAOT at 0.47 μm are close to zero at the level of median values and do not exceed 0.01.The inter-quantile range of the ΔAOT at 0.47 μm is smaller for satellite data as compared to ground-based data. Satellite and ground-based ΔAOT at 0.47 μm are consistent with each other in the diurnal pattern.

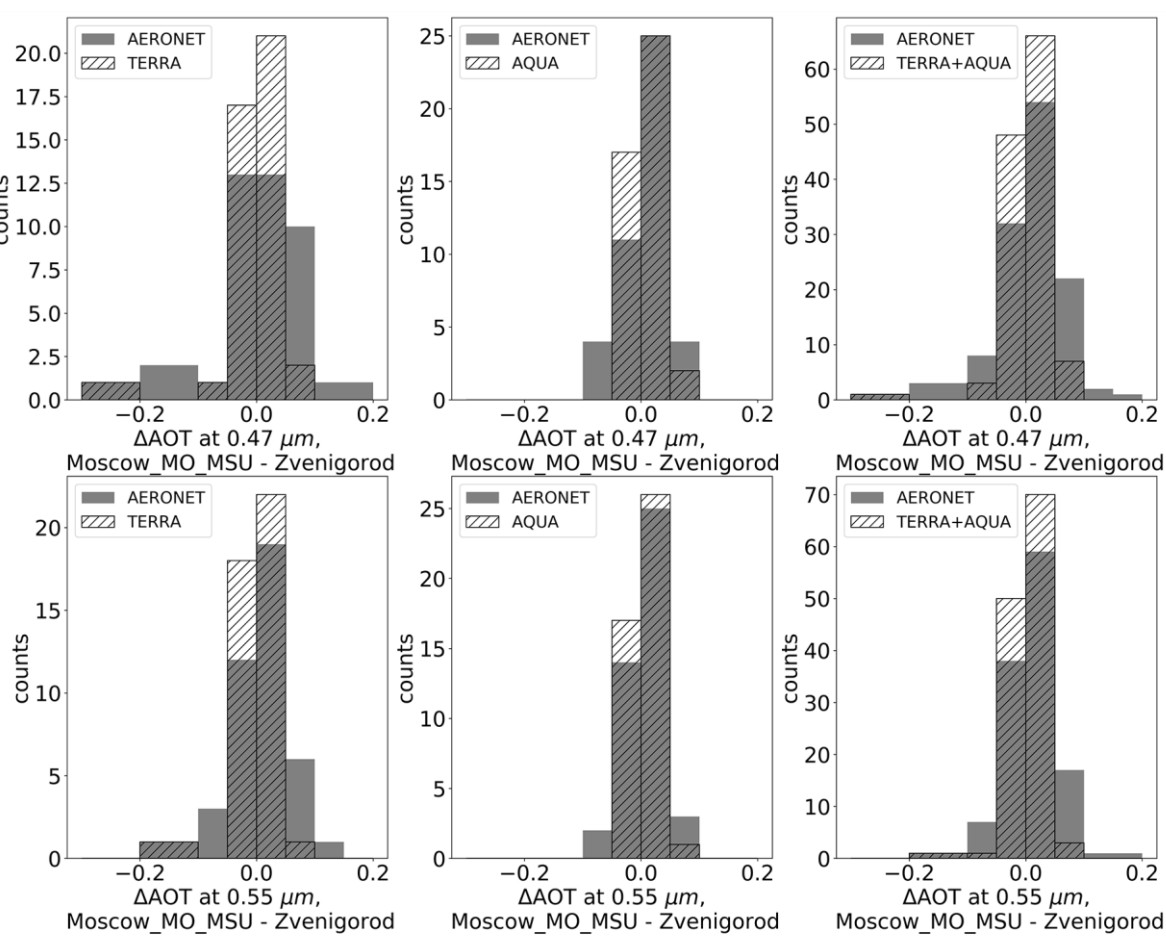


**Figure 6. Frequency distribution of ΔAOT (ΔAOT =AOT$_{Moscow\_MO\_MSU}$ -AOT$_{Zvenigorod}$ ) at 0.47 μm (upper panel) and 0.55 μm (low panel) separately for the Terra (left column) and Aqua (middle column) datasets, and together for the**

data from the two satellites (right column) with frequency distribution for matching ground-based AERONET data, (2006-2017, without the data of 2009 because of technical problems at Zvenigorod AERONET site). Number of satellite and ground-based matchups is 125.

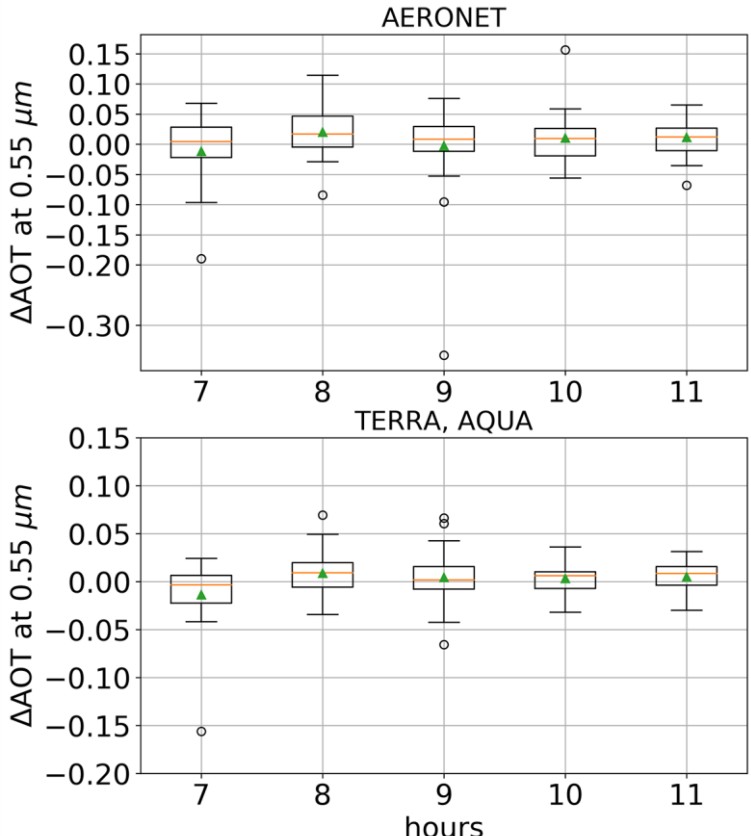

**Figure 7. Daily variations of the ΔAOT at 0.47 μm (ΔAOT =AOT$_{Moscow\_MO\_MSU}$ -AOT$_{Zvenigorod}$ ), UTC time. The median is in the centre, the box is the first (Q1) and the third (Q3) quartiles, the whiskers are Q3 + 1.5 * (Q3-Q1) and Q1 -1.5 * (Q3-Q1), green triangles – means, points – outliers; (2006-2017, without the data of 2009 because of technical problems at Zvenigorod AERONET site). Number of satellite and ground-based matchups is 125.**

For evaluating temporal ΔAOT changes, we analysed variations in annual (warm period) AOT means in Moscow and Zvenigorod. The interannual variations of AOT at 0.55 μm means are shown in Fig. 8 according to the AERONET and MAIAC datasets for the 2006-2017 period. For several years the ΔAOT according to AERONET measurements are statistically significant at 95% confidence level reaching 0.02-0.03 (median value is 0.02), while the MAIAC ΔAOT are

close to zero and not statistically significant for all years. The ΔAOT according to ground-based AERONET observations are positive and higher before 2012. The confidence intervals for the MAIAC data are much larger than the confidence intervals for the AERONET data because of small numbers of satellite matchups.

We excluded AOT for 2009, 2010, and 2013 years in the datasets. The AOT at 0.55 μm was significantly higher in Zvenigorod compared to Moscow in 2009, probably, due to technical problems. Note, that most of the Zvenigorod data during the warm period of 2009 were not included in the previous version 2 AERONET (an email, Alexander Smirnov, personal communication, Aug. 2019). In 2010, the AOT values were strongly affected by extremely high smoke aerosol loading (Chubarova et al., 2012), which was characterized by significant spatial heterogeneity. The data of 2013 year were excluded because of lack of sufficient number of MAIAC observations to obtain robust estimates.

In general, almost for all years we see a tendency of AOT decreasing in Moscow both for the AERONET datasets and satellite retrievals. Similar but less pronounced negative trend of AOT is observed in Zvenigorod. In addition, in the recent years (2013-2017), excluding the 2016 year due to the influence of AOT spatial inhomogeneity of Siberian forest fires, the ΔAOT becomes smaller and, moreover, negative (Fig.8c). We should note that a significant increase in vehicular traffic near the Zvenigorod site, located 150 m away from a road, during past 25 years has resulted in the growth of the surface aerosol air pollution level by about 2-3 times (Kopeikin, et al., 2018), which can lead to the total AOT increase there.

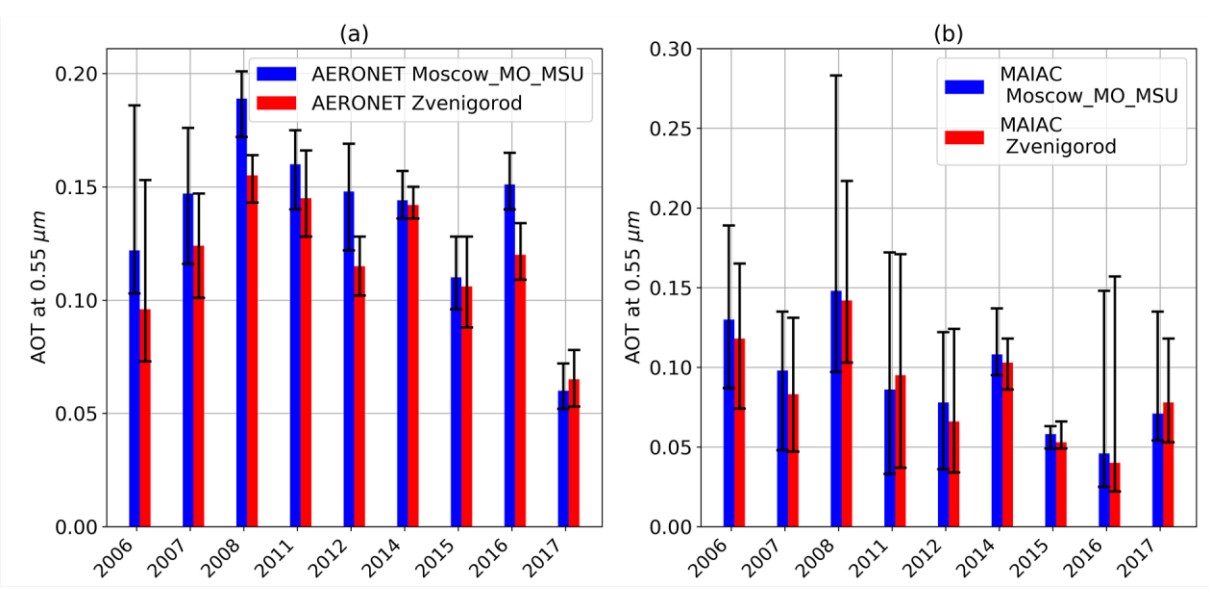

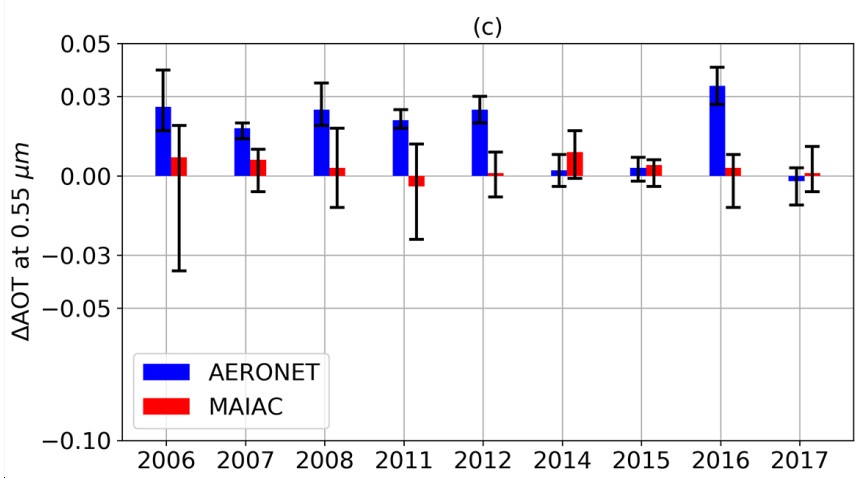

**Figure 8. a) Year-to-year variations of May-September AOT at 0.55 μm medians  (a) - according to all matching AERONET Moscow_MSU_MO and Zvenigorod data (N=1492), (b) - according to the MAIAC data (N= 264), and (c)- ∆AOT according to matching datasets.  Error bars are given at 95% confidence level.**

### 3.4 The spatial AOT distribution over Moscow region and its change in time.

Figure 9 presents the median AOT values for the two time periods (2002–2009 and 2010–2017), which show a decrease in AOT over the territory of "Old" Moscow and an increase over the territory of "New" Moscow. This AOT decrease is consistent with the negative AOT tendency in AOT over Moscow_MSU_MO and Zvenigorod according to AERONET and MAIAC data (see the discussion above).

Spatial changes of AOT over "Old" Moscow and "New" Moscow may be explained by the emissions of urban pollutants - aerosol precursors, and, to some extent, could be associated with the uncertainties in evaluation of the type of underlying surface (for example, the temporal changes in reflectance due to the urban development).

Concerning the possible effect of surface changes, we should note that the MAIAC algorithm provides a dynamic characterization of the surface reflectance properties and spectral ratios required for aerosol retrieval, and should catch temporal surface changes associated with urban development (Lyapustin et al., 2018). In addition, the change in the underlying surface types was analysed using the standard MODIS MCD12C1 Collection 6 product (Majority_Land_Cover_Type_1), which has a spatial resolution of 5 km. The analysis has showed that there was no significant increase in the urban underlying surface over the period 2001–2016.  The number of grid cells occupied by the urban development increased only by 6% over the north of "New" Moscow territory.

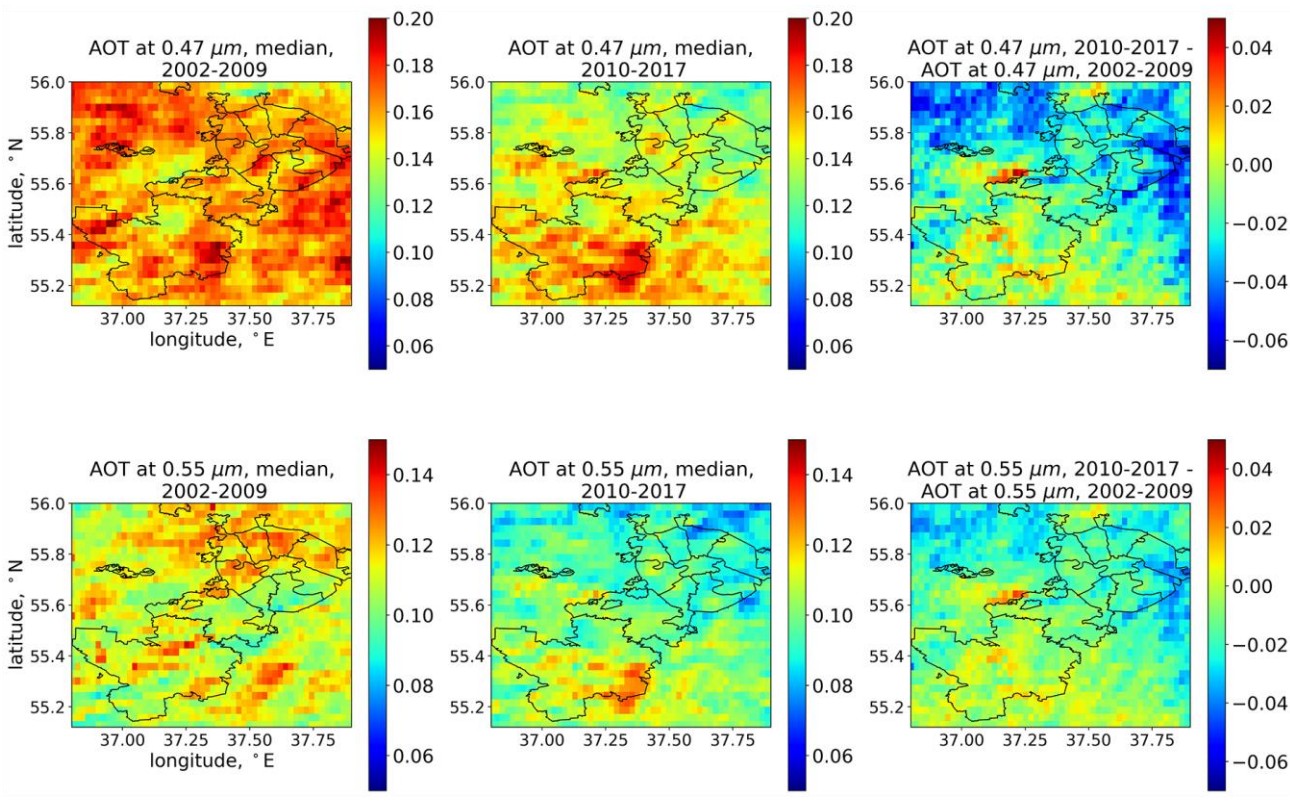

**Figure 9. AOT MAIAC at 0.47 μm and AOT at 0.55 μm median values for the 2002-2009 and 2010-2017 periods and their differences.**

We have also determined the changes in emissions of aerosol precursors for the period 2011-2016 relative to the period 2003-2009 according to the EMEP grid archive (Fig.10). $NO_x$ emissions were characterized by a decrease of about 30% over the territory of Moscow. $NO_x$ emissions from motor vehicles decreased over the considered territory on average by 17%. The decrease of $SO_x$ emissions was on average 14% over the territory of "Old" Moscow and, at the same time, the $SO_x$ emissions increased over the territory of "New" Moscow by about 43%. Emissions of $NH_3$ over the territory of Moscow were increasing, on average by 81%. Emissions of Non-methane volatile organic compound (NMVOC) over the territory of "Old" Moscow were decreasing by about 6%, and at the same time, there was an increase in emissions of NMVOC over the south-west of the considered domain, up to 43%. There was an increase in suspended particles over the territory of "Old" Moscow (+ 16% PM10 and + 6% for PM2.5) and much larger growth in PM (approximately in 2 fold) over the territory of "New" Moscow. However, in recent years there has been a decrease in suspended particles relative to the level of 2010 year.

The obtained results are consistent, for example, with the data in (Chernogaeva et.al., 2019), according to which over the past 10 years, pollutant emissions have decreased in "Old" Moscow, which is caused mainly by environmental regulations

(Kulbachevski et al., 2018), and increased in the Moscow region. Thus, the higher AOT values over the territory of "New" Moscow can be explained by higher aerosol precursors emissions over this area than those over "Old" Moscow.

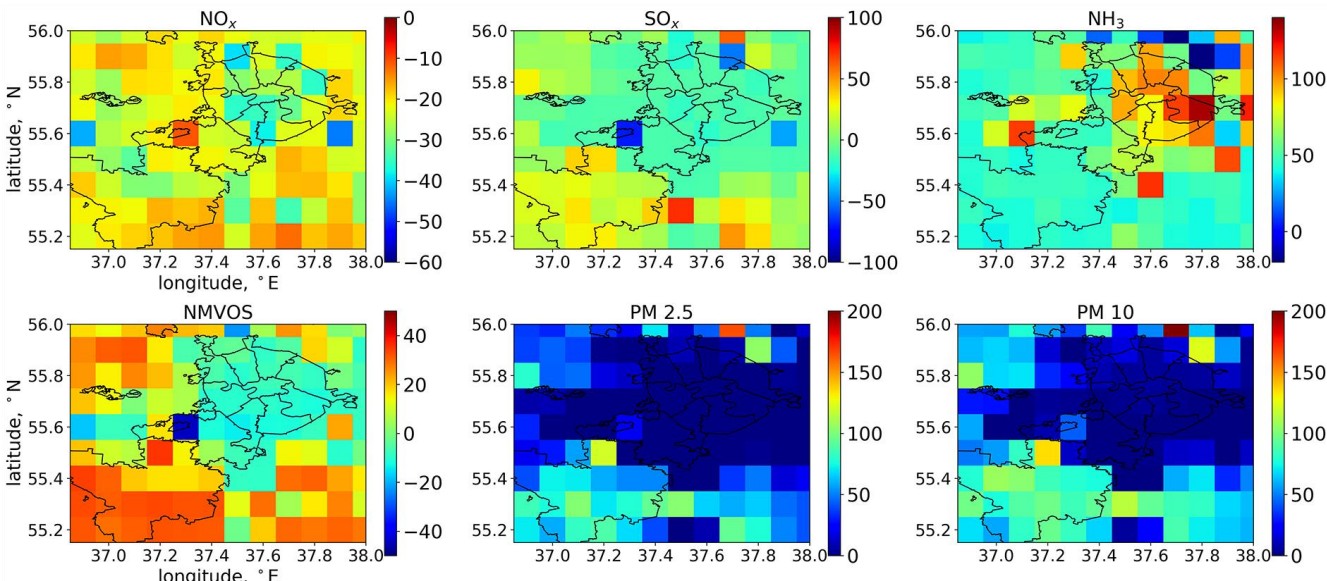

**Figure. 10. Ratio of emissions of gases and particle matter averaged over the 2011-2016 period to the emissions averaged over the 2003-2009 period, in percentages. EMEP dataset (http://www.ceip.at/new_emep-grid/01_grid_data)**

We also applied the quantile analysis to the spatial AOT fields obtained from the MAIAC algorithm separately for the Aqua
and Terra datasets and for both of them. The quantile estimates of AOT over the territory of Moscow region are shown in Fig. 11 and Table 2. In addition to the mentioned elevated mean AOT values over the territory of "New" Moscow, relatively high AOT at 0.47 μm 50% quantile values are observed at the south-western and southern administrative districts of "Old" Moscow (see Fig.1), probably due to highways and industrial enterprises (Fig.11). The spatial changes in AOT over the territory of "Old" Moscow are about 0.03 for wavelength 0.47 μm and 0.55 μm. One can see the most pronounced spatial
differences in AOT at 5% quantile level, where they may reach 0.05-0.06 over several locations in some cases and can be attributed to the stationary sources of aerosol pollution over "Old" Moscow, for example, the areas of building constructions or industrial zones, which can be clearly distinguished in Fig.12. The enhanced AOT over the territory of "New" Moscow are associated with locations of farmlands, which are used in active agricultural activity providing additional aerosol emission. We determined the locations of areas of buildings constructions, industrial zones, farmlands using high resolution
satellite images (WorldView-2, IKONOS).

Table 2 presents mean and maximum values of AOT quantiles for the territories of "Old" and "New" Moscow separately for the Aqua and Terra datasets and for both of them. One can see that over local points the difference between maximum AOT

and mean AOT values comprises about 0.02-0.04 for different quantiles, except 95% quantile, which can be attributed as the local aerosol effect observed in Moscow megacity. Median AOT values according to the Terra dataset are slightly higher (by

0.01-0.02) than the Aqua dataset. The discrepancies in 95% quantile AOT estimates according to these datasets link with the different samples of Terra and Aqua observations.

We also estimated the AOT difference depending on the distance from the city centre. Frequency distribution of AOT at 0.47 μm differences  averaged over the two areas, bounded by circles with a radius of 15 km and 50 km centred in the Moscow city centre consisted of  33% of cases in the range of [-0.02.0] and  60% of cases in the range of [0, 0.02]. This finding is

also consistent with ground-based data.





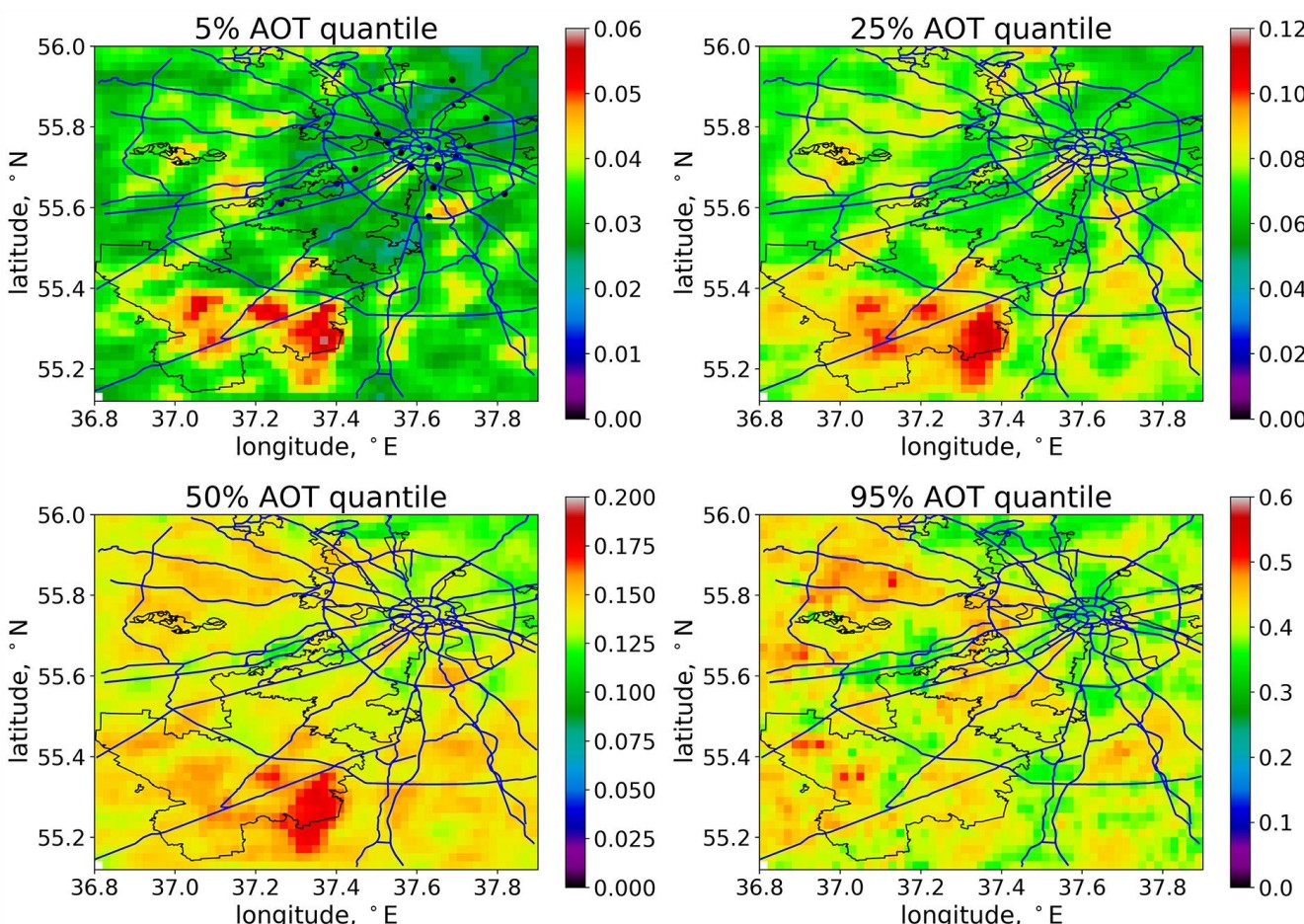

**Figure 11. Quantiles (5%, 25%, 50%, 95%) AOT at 0.47 μm over Moscow megacity, 2001-2017, Aqua and Terra datasets together. Black points in upper left map are thermal power plants according to the «System Operator of the United Power System» data (https://www.so-ups.ru)/. Blue lines are the main highways (data: OpenStreetMap - https://www.openstreetmap.org)**

**Table 2. Mean and maximum of AOT quantiles (5%, 25%, 50%, 95%) over the "Old" Moscow and "New" Moscow territories, 2001-2017.**

| Quantile | "Old" Moscow | | "New" Moscow | |
|---|---|---|---|---|
| | AOT at 0.47 µm (mean/max) | AOT at 0.55 µm (mean/max) | AOT at 0.47 µm (mean/max) | AOT at 0.55 µm (mean/max) |
| Aqua | | | | |
| 5% | 0.03/0.06 | 0.02/0.04 | 0.04/0.06 | 0.02/0.04 |
| 25% | 0.07/0.1 | 0.05/0.07 | 0.08/0.11 | 0.05/0.08 |
| 50% | 0.12/0.15 | 0.08/0.11 | 0.13/0.17 | 0.09/0.12 |
| 95% | 0.34/0.50 | 0.24/0.36 | 0.33/0.52 | 0.23/0.37 |
| Terra | | | | |
| 5% | 0.03/0.04 | 0.02/0.03 | 0.04/0.06 | 0.02/0.04 |
| 25% | 0.07/0.09 | 0.05/0.06 | 0.08/0.12 | 0.06/0.08 |
| 50% | 0.14/0.17 | 0.1/0.11 | 0.15/0.19 | 0.1/0.13 |
| 95% | 0.42/0.52 | 0.3/0.37 | 0.45/0.55 | 0.32/0.39 |
| Aqua and Terra | | | | |
| 5% | 0.03/0.05 | 0.02/0.03 | 0.03/0.06 | 0.02/0.04 |
| 25% | 0.07/0.09 | 0.05/0.06 | 0.08/0.11 | 0.05/0.08 |
| 50% | 0.13/0.16 | 0.09/0.11 | 0.14/0.18 | 0.1/0.12 |
| 95% | 0.39/0.48 | 0.28/0.34 | 0.41/0.51 | 0.29/0.36 |


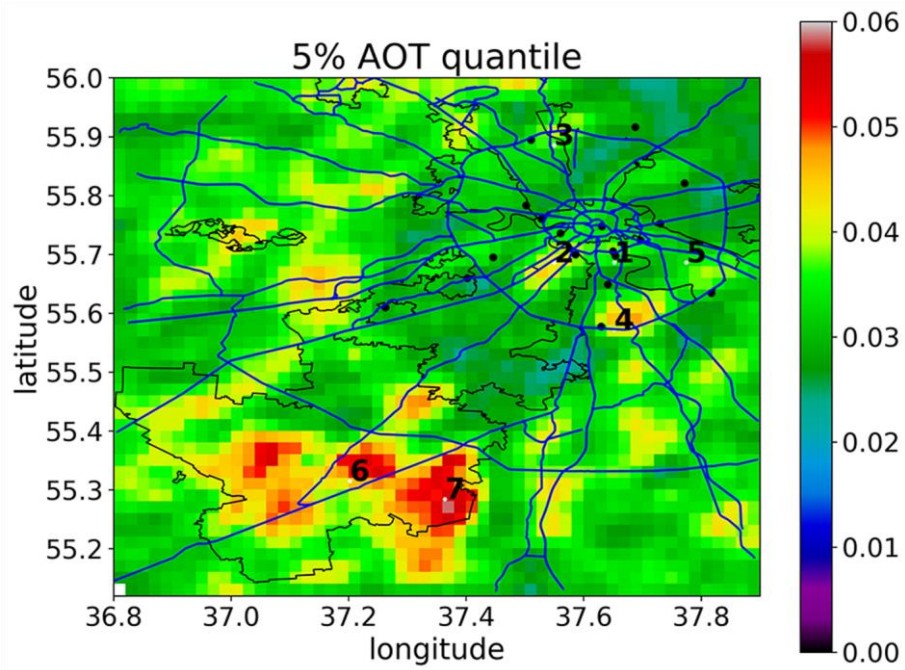

**Figure 12. The 5% quantile of AOT at 0.47 μm, 2001-2017. Points on map: 1, 3, 5 – industrial zones with building construction area, 2, 4 – highways, 6, 7 – farmlands.**


## 4. Discussion and conclusions

The MAIAC AOT (MODIS product MCD19A2) was used for the analysis of the urban aerosol pollution and its dynamics over the Moscow megacity. MAIAC AOT was validated against two AERONET sites located near the centre of Moscow (Moscow_MSU_MO) and in the suburban region (Zvenigorod). The validation showed a good overall agreement between
the ground-based and satellite data, though MAIAC underestimated AOT by 0.05-0.1 for typical conditions (AOT<1). Statistical analysis showed a similar MAIAC AOT performance for the two sites, i.e.  RMSE = 0.1, MAE = 0.07, BIAS = -0.06 for Moscow_MSU_MO  and RMSE = 0.08, MAE = 0.06, BIAS = -0.04 for Zvenigorod. The obtained estimates are consistent with the global MAIAC AOT validation over the land, e.g. RMSE=0.06-0.08 and BIAS= -0.01- -0.03 over the North and South American continents (Lyapustin et al. 2018).

On average, the MAIAC AOT product reproduces the absolute AOT values and the AOT decrease since 2012 observed in the AERONET data, and shows a robust performance in urban environments with higher land surface reflectance. These results are in agreement with other studies, such as Sever et al. (2017) which showed that the pollution from industrial zone could be identified with MAIAC AOT data even over bright semi-deserts of the Dead Sea area.

In high AOT conditions (AOT>1) observed during the Moscow forest and peat fires of 2010, MAIAC showed an overestimation of AOT. This result is in contrast to the typical biomass burning conditions when MAIAC usually underestimates AOT by ~10-20% (e.g., see Lyapustin et al., 2018). MAIAC C6 algorithm lacks a specialized smoke aerosol model with higher absorption and keeps using the regional background aerosol model in cases of detected smoke, which usually has a higher absorption (Dubovik et al., 2002), in particular for the fresh smoke. Atypically, the Moscow 2010 smoke was mostly generated by the slow smouldering peat burning which produces a relatively large particle size and a low absorption (Chubarova et al., 2012, Sayer et al., 2014). The combination of these properties of smoke particles not accounted for in the MAIAC algorithm may have resulted in the observed AOT overestimation. In general, we found that MAIAC smoke detection was a good indicator of forest and peat fires in the Moscow region. Ability of the MAIAC algorithm to confidently capture both fresh and transported smoke in the aerosol type parameter has also been confirmed in Veselovskii et al. (2015).

To evaluate the urban aerosol effect, we analysed the spatial difference between simultaneously measured AOT at Moscow_MSU_MO and at Zvenigorod (ΔAOT=AOT (MOSCOW_MO_MSU) – AOT (ZVENIGOROD)), which was produced from both AERONET and MAIAC datasets. AERONET measurements showed that the annual median ΔAOT varied within -0.002-+0.03 with statistically significant positive bias for most years and the average difference of ~0.02. A similar result was reported for the urban conditions of Warsaw (Zawadzka et al., 2013), where ΔAOT between Warsaw and Belsk was estimated as ~0.02 (at 500 nm) and 0.03 (at 550 nm) according to the AERONET and the standard MODIS aerosol product, respectively. According to Fig. 8, MAIAC also showed a positive ΔAOT ~0.01 between Moscow and Zvenigorod for all years except 2011 (in 2017 both AERONET and MAIAC showed a negative difference) but it was not statistically significant due to higher noise in the MAIAC retrievals compared to the direct AERONET measurements. In comparison, a similar assessment using standard MODIS aerosol algorithm showed ΔAOT=0.03 (Chubarova et al., 2011a). Note, that similar analysis between centre of Berlin city and its suburbs resulted in a much higher ΔAOT=0.08 (Li et al, 2018). Such difference seems to be too high and could be explained by the urban bias of the standard MODIS collection MYD04_3K (3km AOT product) caused by the brighter underlying surface. In previous studies (Remer et al., 2013) MODIS 3 km product based on Dark Target algorithm was shown to have aerosol gradients of better resolution than those obtained from the MODIS 10 km product. However, this product tends to show more noise, especially in urban areas (Munchak et al., 2013). Global validation of MODIS 3 km product exhibits a mean positive bias of 0.06 for Terra and 0.03 for Aqua (Gupta et al., 2018). It was also revealed that MODIS 3 km product overestimates AOT values for Moscow region (Zhdanova, Chubarova, 2018). In recent paper (Jin et al., 2019) an improved AOD retrieval method for 500 m MODIS data has been proposed, which is based on extended surface reflectance estimation scheme and dynamic aerosol models derived from ground-based sun-photometric observations. Its validation with AERONET data showed good results – R = 0.89, while our testing of the MAIAC aerosol product over urban territory of Moscow has revealed correlation coefficient R = 0.97.

Both AERONET and MAIAC show the decreasing trend of the urban aerosol effect (ΔAOT) since 2012, which is consistent with the increase of pollutant emissions over Zvenigorod and their decrease over Moscow during the last years according to the EMEP archive (see Fig. 10).

The analysis of the spatial distribution of MAIAC AOT at 0.47 μm shows higher values over the highways and main roads, building construction areas and over the territory of "New" Moscow at the 5%, 25% and 50% quantile levels with 0.05-0.06 difference against lowest values. The largest local difference in AOT is observed in the clean conditions at 5% quantile. Hence, our results confirm the statement in (Chudnovsky et al., 2013a) that "low pollution days require higher resolution aerosol retrievals to describe spatial AOT heterogeneity in urban environment", which resulted from MAIAC-based study

over the Boston area. The higher AOT over the territory of "New" Moscow can be explained by the increased aerosol precursor emissions from intensive construction and agricultural activities. The difference between the maximum and the mean AOT values for different quantiles, except 95% quantile, within the Moscow region, is about 0.02-0.04 which can be attributed to the local aerosol effects.

Thus, the application of the new MAIAC algorithm provides a reliable instrument for assessing the spatial distribution of

aerosol pollution and allows us to evaluate the level of local aerosol effect of about 0.02-0.04 in visible spectral range over Moscow megacity as well as its temporal dynamics, which has a tendency of AOT decreasing over the "Old" Moscow and increasing over the "New" Moscow territories.

In this research we have verified the MAIAC algorithm data against ground-based data and obtained spatial and temporal variability of AOT MAIAC retrievals over Moscow region for evaluating aerosol pollution. Future studies focused on

influence of different meteorological conditions on AOT MAIAC retrievals will be valuable for detection events of the extreme urban aerosol pollution and further MAIAC product validation.

**Data availability**

The MODIS product data - MCD19A2 Collection 6 (MAIAC aerosol product) and MCD12C1 Collection 6 product were

obtained from https://search.earthdata.nasa.gov/search. Grid archive of aerosol precursor gases emissions and particulate matter concentrations is available at http://www.ceip.at/new_emep-grid/01_grid_data. The AERONET version 3 data at the Moscow_MO_MSU and Zvenigorod sites are available from the AERONET data repository at https://aeronet.gsfc.nasa.gov.

**Author contribution**

EYuZ and NYC designed the study and wrote the paper with essential contributions from AIL. EYuZ was responsible for data collection and visualization. Data analysis was performed by EYuZ and NYC.

**Competing interests**

The authors declare that they have no conflict of interest.

**Acknowledgements**

This work is supported by the Russian Science Foundation under grant # 18-17-00149. The work of A. Lyapustin was supported by the NASA Science of Terra, Aqua, SNPP (17-TASNPP17-0116; solicitation NNH17ZDA001N-TASNPP). We thank the RAS Zvenigorod Scientific Station stuff for their efforts in establishing and maintaining Zvenigorod AERONET site.

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
