# Peer review of "Assessment of urban aerosol pollution over Moscow megacity by MAIAC aerosol product."

_Atmospheric Measurement Techniques, 2019_

## Referee Comment (RC1) · Anonymous Referee #3 · 5 Nov 2019

In this paper, the authors considered the distribution of aerosol optical thickness (AOT) with a spatial resolution of 1 km over Moscow megacity using MAIAC aerosol product based on MODIS satellite data (for the warm period of year (May-September)). The subject area of the study is urgent because an efficient aerosol retrieval under heavy loading conditions is critical and it can be useful for investigations on regional climate change, air pollution control, and aerosol.

General comments

1. For a more detailed description of the spatial distribution of aerosol over Moscow megacity the authors use the MAIAC aerosol product with a spatial resolution of 1 km. It is reasonable to add a section (or subsection), comparing the obtained results not only with data from ground-based AERONET observations at Moscow_MSU_MO_site

and Zvenigorod site (Zvenigorod scientific station of Institute of Atmospheric Physics RAS), but also with data of standard MODIS collection MYDD04_3K (3K AOT product).

2. It is not quite clear why the authors included in the paper the results concerning the distribution of ΔÐŘÐđÐć for different morning hours (Figures 7-8). Is this still another aspect associated with validation? Why, although presenting data exclusively for morning hours, the authors nonetheless say about diurnal variations of ΔÐŘÐđÐć?

3. It is useful to turn attention to the paper by Jin et al., Retrieval of 500 m Aerosol Optical Depths from MODIS Measurements over Urban Surfaces under Heavy Aerosol Loading Conditions in Winter, Remote Sens. 2019, 11, 2218; doi:10.3390/rs11192218. That paper appeared after E. Zhdanova and coauthors had already submitted their research for publication in AMT. However, at this stage it makes sense to compare the results, obtained by the authors, with data, presented by Jin et al., 2019

Minor comments

1. Line numbers 124-125: "... MAIAC AOT data were spatially averaged with a 5-km circle 125 centred at the Moscow_MSU_MO and Zvenigorod sites...". Why circle with diameter (radius?) of 5 km is chosen?

2. Line number 136: "... Statistical estimates of the quality of the AOT...". Caption of Table 1 indicates precisely what characteristics are considered by the authors. It would be better to move them to the text of the paper because the indicated abbreviations are also used below (see, e.g., line number 364).

3. Figure 2. Information on fitting equation, correlation coefficient, root-mean-square and number of retrieval should be added in the field of the figure.

4. Figure 4 and comments. In section 2 (line numbers 87-88) it is indicated that "MAIAC uses 8 different regional aerosol models tuned to the AERONET...". What the data in Fig. 4b, accompanied by the comments "MAIAC", and indication that "MAIAC is regional model", correspond to, in this case?

5. It makes sense to work on the style of the presentation. For example, within one paragraph the authors write "One can see......" (line numbers 327, 330), "We can see......" (line number 333), etc.

6. The reference

Sever, L., Alpert, P., Lyapustin, A., Wang, Y. and Chudnovsky, A.: An example of aerosol pattern variability over bright surface using high resolution MODIS MAIAC: The eastern and western areas of the Dead Sea and environs, Atmospheric Environment, 165, 359–369, doi:10.1016/j.atmosenv.2017.06.047, 2017.

is repeated twice.

In my opinion, the article contains useful information and can be published after revision.

Please also note the supplement to this comment:
https://www.atmos-meas-tech-discuss.net/amt-2019-325/amt-2019-325-RC1-supplement.pdf

---

## Referee Comment (RC2) · Anonymous Referee #1 · 10 Nov 2019

The submitted manuscript entitled "Assessment of urban aerosol pollution over Moscow megacity by MAIAC aerosol product" presents an important application of MAIAC AOD retrievals over urban mega-city environment. I would like to recommend the publication however after the paper major revision. For my opinion, authors need to make additional effort to improve the manuscript before its acceptance. Below are my suggestions.

General comments:

1. I think it will be great if authors provide spatial and seasonal/temporal average pattern of AOD. Furthermore, on AOD images all important geographic locations must be shown: suburban cities, city center, etc. Readers are not familiar with Moscow geography and it is difficult to follow authors results. Also at the beginning authors

need to explain the differences between New Moscow and Old Moscow under section of "Study Area". And where are these regions on the map? Otherwise i discovered the differences in pollution pattern between both parts only at the end of a paper. Introduction should be devoted to the previous studies done in the subject that are the most relevant to the authors study rather than to study area explanation that should be only briefly explained.

2. My additional comments relate to the analyses of AOD percentiles (Figure 12- which is interesting). Without a general/AOD average maps, I find it difficult to analyze results of AOD lower/upper percentiles. I also think that these analyses are speculative and must be very carefully presented, more as authors interpretation, as a "hint to local pollution", hint to regional, etc with references as done in Discussion. May be including this figure in Discussion section would be better?. And comparison with ground confirmation of these results? With some critical statements of of these results.

3. May be some figures can be removed as it reduces the paper clarity. Some figures are not explained and not well presented (details are below).

Specific/minor comments:

Introduction:

Additional literature search is required. For example: - Line 37-39: Elevated aerosol loading is generally correlated with suspended particulate matter associated with the poor air quality (van Donkelaar et al., 2015, Beloconi et al., 2018).Authors need to add additional citations that originally investigate the subject. For example, the correlation between particulate matter concentrations and AOD is not a new subject and was widely discussed. As pointed out in Hoff and Christopher 2009 (review article), different geographic locations exhibit different correlations.

Wang and Christopher 2003: Intercomparison between satellite-derived aerosol optical thickness and PM2.5 mass: Implications for air quality studies

Hoff and Christopher 2009: Remote Sensing of Particulate Pollution from Space: Have We Reached the Promised Land? JAWMA 59(6), 645-675.

Look at Figure 3 in Chudnovsky et al. 2012 "Prediction of daily fine particulate matter concentrations using aerosol optical depth retrievals from the Geostationary Operational Environmental Satellite (GOES)" JAWMA V(62)

- The use of AOD in atmospheric application is excellently presented by Kaufman et al. 2002: "A satellite view of aerosols in the climate system" published in Nature.

- Line 44: Authors stated "recent studies" Although I do not find citations to 2011 or 2013 as recent studies. I searched what was done with MAIAC recently- and perhaps can be relevant- up to authors decision of course: Barnaba et al. 2018: Satellite-based view of the aerosol spatial and temporal variability in the Córdoba region (Argentina) using over ten years of high-resolution data, ISPRS Journal of Photogrammetry and Remote Sensing. And more publications can be found.

-Line 74: "against the high-quality AERONET measurement". I would avoid such a strong statement as "high-quality" Sometimes even AERONET provide biased measurements. I would suggest "against ground-based AOD measurements".

Methods: - Methodology section and data sets-all is mixed up. One needs to dig the information. Please reorganize to sub-paragraphs MAIAC AOD, AERONET data, gaseous pollution data, study area, etc. The same is for results section.

Results: 1. Figure 2: authors need to provide equation for both plots, slope, intercept, r, and explain high residuals on both plots, what are possible causes.

2. I do not understand Figure 5- it says correlation, but I do not see correlation coefficient, I do not see any pattern except of lack of it. I see a scatter plot with zero correlation. What authors wanted to present? I get puzzled.

3. Figure 10: I do not understand what median AOD maps present? Why authors can't present average AOD values instead? Please justify your selection.

4. Figures 6 and 7 are not explained. Please provide explanations to your results.

Discussion:

Authors need to provide discussion on points that overestimated and underestimated by MAIAC AOD retrieval at least by showing what are meteorological conditions that favor these results. I mean- analyses of residuals (from figure 2).

Authors also need to state the limitations of thier results and future directions in one short paragraph.

Good Luck!

---

## Author Comment (AC1) · 29 Dec 2019

*(1) comments from Referees (are marked by italics),* (2) author's response (plain text), (3) author's changes in manuscript (are marked by yellow color).

*General comments:*

*1.   I think it will be great if authors provide spatial and seasonal/temporal average pattern of AOD.*

Long-term AERONET measurements at Moscow (Moscow_MO_MSU site) demonstrate that seasonal variations of AOT are noticeable with maximum in April and July (median AOT at 0.5 μm are equal 0.22-0.21) and minimum in December and January (median AOT at 0.5 μm is equal 0.07).  There are a few previous publications concerning AOT seasonal and temporal variations (for example, Chubarova et al. 2011, Chubarova et al., 2016). We added the discussion about AOT changes in the manuscript in subsection 2.3 (see below). In present research, we considered only warm period of year (May-September).  In this period of year, AOT variations are not large (~ 0.15-0.21).

Spatial variations of AOT are shown at Fig.12 (now Fig.11) in the first version of the manuscript. Our main objective is to discover spatial structure of AOT, to reveal possible local pollution based on MAIAC product over Moscow region.

Changes in manuscript: subsection **2.3 AERONET data**

Long-term measurements at the Moscow_MSU_MO have revealed noticeable seasonal changes in AOT with maximum in April and July with median AOT at 0.5 μm  of about 0.22, and minimum in December and January with AOT at 0.5 μm of 0.07 (Chubarova et al. 2011b, Chubarova et al., 2016). However, in this study we focused on snow-free period (May-September), during this period of year AOT variations are not large (~ 0.15-0.21).

*Furthermore, on AOD images all important geographic locations must be shown: suburban cities, city center, etc. Readers are not familiar with Moscow geography and it is difficult to follow authors results. Also at the beginning need to explain the differences between New Moscow and Old Moscow under section of "Study Area". And where are these regions on the map? Otherwise i discovered the differences in pollution pattern between both parts only at the end of a paper. Introduction should be devoted to the previous studies done in the subject that are the most relevant to the authors study rather than to study area explanation that should be only briefly explained.*

We agree that an additional information about Moscow geography is needed. We updated the Fig.1 providing satellite image and administrate division of Moscow.  We moved from Introduction to special subsection information about study area. We called "Old" Moscow is the city territory before 2012 year.  In 2012, the Moscow megacity has expanded to the south-west and we called this new territory as "New" Moscow. "Old" Moscow is marked by yellow color and "New" Moscow is marked by green color in Fig.1. We also modified the Introduction, including more references. Please, see the details of changes below.

Changes in manuscript:

Fig. 1 is updated.

a)

[revised manuscript text omitted]

*2. My additional comments relate to the analyses of AOD percentiles (Figure 12- which is interesting). Without a general/AOD average maps, I find it difficult to analyze results of AOD lower/upper percentiles. I also think that these analyses are speculative and must be very carefully presented, more as authors interpretation, as a "hint to local pollution", hint to*

*regional, etc with references as done in Discussion. May be including this figure in Discussion section would be better? And comparison with ground confirmation of these results? With some critical statements of these results.*

In the analysis we decided not to use average AOT values, but to focus on quantile AOT analysis to avoid impact of forest and peat fires causing non-periodic strong AOT inhomogeneity. The events of big forest and peat fires strongly influence the mean AOT values. We think that using of median values is a robust way to show the AOT spatial distribution. We added lines of main roads and highways in Figure 12 (old number, now Figure 11) and now the links of enhanced AOT values with urban emissions (roads, power stations) are seen much better. We also updated Figure 12 (now Figure 11). We found that on the 5% quantile map the maximum of AOT corresponding to large areas of building construction or industry zone and farmlands. We added an additional Figure, where several points of local pollution were shown to associate with location of anthropogenic objects (road, building construction areas). We found these points by visual examination of high resolution satellite images.

Changes in manuscript:

[revised manuscript text omitted]

*3. May be some figures can be removed as it reduces the paper clarity. Some figures are not explained and not well presented (details are below).*

We removed fig. 7, please see changes below.

*Specific/minor comments:*

*1. Introduction:*

*Additional literature search is required. For example: - Line 37-39: Elevated aerosol loading is generally correlated with suspended particulate matter associated with the poor air quality (van Donkelaar et al., 2015, Beloconi et al., 2018). Authors need to add additional citations that originally investigate the subject. For example, the correlation between particulate matter concentrations and AOD is not a new subject and was widely discussed. As pointed out in Hoff and Christopher 2009 (review article), different geographic locations exhibit different correlations.*

*Look at Figure 3 in Chudnovsky et al. 2012 "Prediction of daily fine particulate matter concentrations using aerosol optical depth retrievals from the Geostationary Operational Environmental Satellite (GOES)" JAWMA V(62) - The use of AOD in atmospheric application is excellently presented by Kaufman et al. 2002: "A satellite view of aerosols in the climate system" published in Nature.*

*- Line 44: Authors stated "recent studies" Although I do not find citations to 2011 or 2013 as recent studies. I searched what was done with MAIAC recently- and perhaps can be relevant- up to authors decision of course: Barnaba et al. 2018: Satellite-based view of the aerosol spatial and temporal variability in the Córdoba region (Argentina) using over ten years of high-resolution data, ISPRS Journal of Photogrammetry and Remote Sensing. And more publications can be found.*

Thank you! We added additional discussion in the Introduction, please see edited text of Introduction above. Since the analysis of relationship between particulate matter and AOT is not our main scope we paid on this subject not much attention.

*-Line 74: "against the high-quality AERONET measurement". I would avoid such a strong statement as "high-quality" Sometimes even AERONET provide biased measurements. I would suggest "against ground-based AOD measurements".*

We did this correction, but we should mention that we use additional cloud filtering by visual observations of cloudiness for AOT AERONET data version 3, so the used data is really tend to be high-quality.

*Methods: - Methodology section and data sets-all is mixed up. One needs to dig the information. Please reorganize to sub-paragraphs MAIAC AOD, AERONET data, gaseous pollution data, study area, etc. The same is for results section.*

We divided the Section 2. The study area, datasets and methodology into subsections : 2.1 The study area, 2.2. MAIAC data, 2.3 AERONET data, 2.4 EMEP data.
The section Results is not changed and consists of several subsections.

*Results: 1. Figure 2: authors need to provide equation for both plots, slope, intercept, r, and explain high residuals on both plots, what are possible causes.*

Yes, of course, now we provided fitting equations. We updated figure 2. One can see that the correlation between AOT MAIAC and AERONET is high for Moscow_MO_MSU site, R =0.91-0.97.

Changes in manuscript:

However, the correlation between the AOT MAIAC retrievals and AERONET data is high. Slopes of regressions lines are higher at the Moscow_MO_MSU site than that at Zvenigorod, since at Zvenigorod site high aerosol loading due to forest and peatbog fires has not been included in the sample.

[Figure]

**Figure 2. Correlations between MAIAC AOT at 0.47 μm and AERONET AOT at 0.47 μm for Moscow_MSU_MO and Zvenigorod AERONET sites for Terra, Aqua and their joint overpasses within 1 hour (Aqua/Terra).**

**Comment: the absence of high AOT values at Zvenigorod site is explained by technical problems with the instrument and the absence of the AERONET data at level 2 version 3 in 2010, when intensive forest fires took place.**

*2. I do not understand Figure 5- it says correlation, but I do not see correlation coefficient, I do not see any pattern except of lack of it. I see a scatter plot with zero correlation. What authors wanted to present? I get puzzled.*

We made changes in the manuscript: Fig.5 shows a relationship between dAOT from MAIAC and from hourly-averaged AERONET data. The ΔAOT values obtained from both ground-based and satellite data lie within the range of -0.1 ... 0.1. It should be noted that the ΔAOT between Moscow_MSU_MO and Zvenigorod based on satellite and ground-based data generally correspond to each other.

The caption of Fig.5 has been changed.

[Figure]

a                                                                 b

**Figure 5. (a) Relationship between dAOT at 0.47μm (ΔAOT=AOT$_{Moscow\_MO\_MSU}$ - AOT$_{Zvenigorod}$) obtained from the satellite and ground-based data; (b) ΔAOT at 0.47μm as a function of AOT at 0.47μm obtained from Moscow_MSU_MO dataset.**

*3. Figure 10: I do not understand what median AOD maps present? Why authors can't present average AOD values instead? Please justify your selection.*

We chose median values to show robust unbiased AOT spatial distribution. We do not use mean AOT values to avoid impact of forest and peat fires causing non-periodic strong AOT inhomogeneity which significantly influence on the average estimates.

*4. Figures 6 and 7 are not explained. Please provide explanations to your results.*

We decided to remove Fig.7, because it repeat in some extent Fig.8 and changed the text as following:

For characterizing variations in ΔAOT we analysed frequency distributions according to ground-based and satellite data. In general, polar orbiting satellites demonstrate similar daily average

==AOT independent of morning or afternoon orbits (Kaufman et al., 2000).== However, we calculated ∆AOT separately for Terra and Aqua datasets for evaluating to some extent diurnal (==in the morning and noon hours==) variability of ∆AOT. Frequency distributions of ∆AOT at 0.47 and 0.55 µm separately for the Terra and Aqua data, and together for the data from the two satellites are shown in Fig.6. The highest repeatability of ∆AOT is in the range of 0-0.05. For the Aqua AOT retrievals, which are closer to noon, the predominance of positive ∆AOT is more pronounced. ==Fig. 6 also shows a large negative ∆AOT in cases of Terra measurements in our sample.== In overall, the ∆AOT at 0.47 values lie within the [0, 0.05] bin in 57% of cases for the Aqua and in 50% - for the Terra datasets.

The diurnal variations of the ∆AOT according to satellite and ground-based data are also shown in ==Fig.7.== The MAIAC ∆AOT at 0.47 µm are close to zero at the level of median values and do not exceed 0.01.The inter-quantile range of the ∆AOT at 0.47 µm is smaller for satellite data as compared to ground-based data. Satellite and ground-based ∆AOT at 0.47 µm are consistent with each other in the diurnal pattern.

[Figure]

**Figure 6. Frequency distribution of ∆AOT (∆AOT =AOT$_{Moscow\_MO\_MSU}$ -AOT$_{Zvenigorod}$ ) at 0.47 µm (upper) and 0.55 µm (low) separately for the Terra (left column) and Aqua (middle column) datasets, and together for the data from the two satellites (right column) with frequency distribution for matching ground-based AERONET data, (2006-2017, without the data of 2009 because of technical problems at Zvenigorod AERONET site). Number of satellite and ground-based matchups is 125.**

[Figure]

**Figure 7. Daily variations of the ΔAOT at 0.47 µm (ΔAOT =AOT$_{Moscow\_MO\_MSU}$ - AOT$_{Zvenigorod}$ ), UTC time. The median is in the centre, the box is the first (Q1) and the third (Q3) quartiles, the whiskers are Q3 + 1.5 * (Q3-Q1) and Q1 -1.5 * (Q3-Q1), green triangles – means, points – outliers; (2006-2017, without the data of 2009 because of technical problems at Zvenigorod AERONET site). Number of satellite and ground-based matchups is 125.**

*Discussion:*
*Authors need to provide discussion on points that overestimated and underestimated by MAIAC AOD retrieval at least by showing what are meteorological conditions that favor these results. I mean- analyses of residuals (from figure 2).*

In this research, we paid main attention to the analysis of aerosol model used in MAIAC AOT algorithm. We showed that AOT MAIAC were overestimated for smoke conditions with AOT>1 due to spatial and temporal variability of smoke properties, which can be various in different geographical regions. Cases studies of influence of meteorological conditions on AOT MAIAC product is the issue of our future research, which is now mentioned in the text.

*Authors also need to state the limitations of their results and future directions in one short paragraph.*

We added concluding remarks, please see below

Changes in manuscript:

Thus, the application of the new MAIAC algorithm provides a reliable instrument for assessing the spatial distribution of aerosol pollution and allows us to evaluate the level of local aerosol effect of about 0.02-0.04 in visible spectral range over Moscow megacity as well as its temporal dynamics, which has a tendency of AOT decreasing over the "Old" Moscow and increasing over the "New" Moscow territories.

In this research we have verified the MAIAC algorithm data against ground-based data and obtained spatial and temporal variability of AOT MAIAC retrievals over Moscow region for evaluating aerosol pollution. Future studies focused on influence of different meteorological conditions on AOT MAIAC retrievals will be valuable for detection events of the extreme urban aerosol pollution and further MAIAC product validation.

All changes in the manuscript are marked by yellow color.

---

## Author Comment (AC2) · 29 Dec 2019

*(1) comments from Referees (are marked by italics),* (2) author's response (plain text), (3) author's changes in manuscript (are marked by yellow color).

*General comments*

*1. For a more detailed description of the spatial distribution of aerosol over Moscow megacity the authors use the MAIAC aerosol product with a spatial resolution of 1 km. It is reasonable to add a section (or subsection), comparing the obtained results not only with data from ground-based AERONET observations at Moscow_MSU_MO_site and Zvenigorod site (Zvenigorod scientific station of Institute of Atmospheric Physics RAS), but also with data of standard MODIS collection MYDD04_3K (3K AOT product).*

Our main task was to try to identify local aerosol pollution by satellite measurements in urban environment. For this purpose we test MAIAIC aerosol product. The previous research was shown that MODIS 3 km product provides higher estimates of AOT on the cite center of Moscow. We added additional information about MODIS 3 km product in the manuscript:

changes in manuscript:

In Discussion it was added:

In previous studies (Remer et al., 2013) MODIS 3 km product based on Dark Target algorithm was shown to have aerosol gradients of better resolution than those obtained from the MODIS 10 km product. However, this product tends to show more noise, especially in urban areas (Munchak et al., 2013). Global validation of MODIS 3 km product exhibits a mean positive bias of 0.06 for Terra and 0.03 for Aqua (Gupta et al., 2018). It was also revealed that that MODIS 3 km product overestimates AOT values for Moscow region (Zhdanova, Chubarova, 2018).

Added references:
Munchak, L. A. L.: MODIS 3 Km Aerosol Product: Applications over Land in an Urban/suburban Region, Atmospheric Measurement Techniques, 1747–1759, doi:10.5194/amt-6-1747-2013, http://dx.doi.org/10.5194/amt-6-1747-2013, 2013.
Remer, L. A., Mattoo, S., Levy, R. C. and Munchak, L. A.: MODIS 3 km aerosol product: algorithm and global perspective, Atmospheric Measurement Techniques, 6(7), 1829–1844, doi:https://doi.org/10.5194/amt-6-1829-2013, 2013.

Gupta, P., Remer, L. A., Levy, R. C. and Mattoo, S.: Validation of MODIS 3 km land aerosol optical depth from NASA's EOS Terra and Aqua missions, Atmospheric Measurement Techniques, 11(5), 3145–3159, doi:https://doi.org/10.5194/amt-11-3145-2018, 2018.

*2. It is not quite clear why the authors included in the paper the results concerning the distribution of dAOT for different morning hours (Figures 7-8). Is this still another aspect associated with validation? Why, although presenting data exclusively for morning hours, the authors nonetheless say about diurnal variations of dAOT?*

It was interesting to see if there is any change in diurnal (we mean variations in morning and noon hours) change in dAOT using MAIAC data. But we have obtained the absence of significant dAOT changes in morning and noon hours. We decided to remove Fig.7, because it repeats to some extent Fig.8. The changed text is as following:

"For characterizing variations in $\Delta$AOT we analysed frequency distributions according to ground-based and satellite data. In general, polar orbiting satellites demonstrate similar daily average AOT independent of morning or afternoon orbits (Kaufman et al., 2000). However, we calculated $\Delta$AOT separately for Terra and Aqua datasets for evaluating possible diurnal (in the morning and noon hours) variability of $\Delta$AOT. Frequency distributions of $\Delta$AOT at 0.47 and 0.55 μm separately for the Terra and Aqua data, and together for the data from the two satellites are shown in Fig.6. The highest repeatability of $\Delta$AOT is in the range of 0-0.05. For the Aqua AOT retrievals, which are closer to noon, the predominance of positive $\Delta$AOT is more pronounced. Fig. 6 also shows a large negative $\Delta$AOT in cases of Terra measurements in our sample. In overall, the $\Delta$AOT at 0.47 values lie within the [0, 0.05] bin in 57% of cases for the Aqua and in 50% - for the Terra datasets.

The diurnal variations of the $\Delta$AOT according to satellite and ground-based data are also shown in Fig.7. The MAIAC $\Delta$AOT at 0.47 μm are close to zero at the level of median values and do not exceed 0.01.The inter-quantile range of the $\Delta$AOT at 0.47 μm is smaller for satellite data as compared to ground-based data. Satellite and ground-based $\Delta$AOT at 0.47 μm are consistent with each other in the diurnal pattern."

[Figure]

**Figure 6. Frequency distribution of $\Delta$AOT ($\Delta$AOT =AOT$_{Moscow\_MO\_MSU}$ -AOT$_{Zvenigorod}$ ) at 0.47 µm (upper) and 0.55 µm (low) separately for the Terra (left column) and Aqua (middle column) datasets, and together for the data from the two satellites (right column) with frequency distribution for matching ground-based AERONET data, (2006-2017, without the data of 2009 because of technical problems at Zvenigorod AERONET site). Number of satellite and ground-based matchups is 125.**

[Figure]

**Figure 7. Daily variations of the ΔAOT at 0.47 μm (ΔAOT =AOT_Moscow_MO_MSU -AOT_Zvenigorod ), UTC time. The median is in the centre, the box is the first (Q1) and the third (Q3) quartiles, the whiskers are Q3 + 1.5 * (Q3-Q1) and Q1 -1.5 * (Q3-Q1), green triangles – means, points – outliers; (2006-2017, without the data of 2009 because of technical problems at Zvenigorod AERONET site). Number of satellite and ground-based matchups is 125.**

*3. It is useful to turn attention to the paper by Jin et al., Retrieval of 500 m Aerosol Optical Depths from MODIS Measurements over Urban Surfaces under Heavy Aerosol Loading Conditions in Winter, Remote Sens. 2019, 11, 2218; doi:10.3390/rs11192218. That paper appeared after E. Zhdanova and coauthors had already submitted their research for publication in AMT. However, at this stage it makes sense to compare the results, obtained by the authors, with data, presented by Jin et al., 2019*

Thank you. We added this paper in the analysis.

In recent paper (Jin et al., 2019) an improved AOD retrieval method for 500 m MODIS data has been proposed, which is based on extended surface reflectance estimation scheme and dynamic aerosol models derived from ground-based sun-photometric observations. Its validation with

AERONET data showed good results – R = 0.89, while our testing of the MAIAC aerosol product over urban territory of Moscow has revealed correlation coefficient R = 0.97.

*Jin, S., Ma, Y., Zhang, M., Gong, W., Dubovik, O., Liu, B., Shi, Y. and Yang, C.: Retrieval of 500 m Aerosol Optical Depths from MODIS Measurements over Urban Surfaces under Heavy Aerosol Loading Conditions in Winter, Remote Sensing, 11(19), 2218, doi:10.3390/rs11192218, 2019.*

*Minor comments*

*1. Line numbers 124-125: ": : : MAIAC AOT data were spatially averaged with a 5-km circle 125 centred at the Moscow_MSU_MO and Zvenigorod sites: : :". Why circle with diameter (radius?) of 5 km is chosen?*

Usually, 27 km radius is chosen for satellite validation of AOT, but we used 5km radius to catch the possible features of the underlying urban and suburban surfaces.

*2. Line number 136: ": : : Statistical estimates of the quality of the AOT: : :". Caption of Table 1 indicates precisely what characteristics are considered by the authors. It would be better to move them to the text of the paper because the indicated abbreviations are also used below (see, e.g., line number 364).*

We changed the text:

Statistical estimates (RMSE - root mean square error, MAE - mean absolute error, BIAS - mean error) of the quality of the AOT at 0.47 µm retrievals relative to the ground-based AERONET data are presented in Table 1.

*3. Figure 2. Information on fitting equation, correlation coefficient, root-mean-square and number of retrieval should be added in the field of the figure.*

Fitting equations, correlation coefficients are added on figures, RMSE and Number of retrievals are presented in Table 2.

[Figure]

| MOSCOW_MO_MSU | ZVENIGOROD |
|---|---|

**Figure 2. Correlations between MAIAC AOT at 0.47 μm and AERONET AOT at 0.47 μm for Moscow_MSU_MO and Zvenigorod AERONET sites for Terra, Aqua and their joint overpasses within 1 hour (Aqua/Terra).**

**Comment: the absence of high AOT values at Zvenigorod site is explained by technical problems with the instrument and the absence of the AERONET data at level 2 version 3 in 2010, when intensive forest fires took place.**

*4. Figure 4 and comments. In section 2 (line numbers 87-88) it is indicated that "MAIAC uses 8 different regional aerosol models tuned to the AERONET: : :". What the data in Fig. 4b, accompanied by the comments "MAIAC", and indication that "MAIAC is regional model", correspond to, in this case?*

The geographic distribution of regional background aerosol models over land used in MAIAC processing is shown in Fig. 4 from (Lyapustin, A., Wang, Y., Korkin, S. and Huang, D.: MODIS Collection 6 MAIAC algorithm, Atmospheric Measurement Techniques, 11(10), 5741–5765, doi:https://doi.org/10.5194/amt-11-5741-2018, 2018.), please, see below. Each geographical location has one predefined aerosol model. Aerosol model number 1 is used for Moscow region. Additionally smoke/dust tests are applied.

[Figure]

**Figure 4.** Map of background regional aerosol models specified in Table 1. The transparent yellow shape approximates the dust regions.

Changes in manuscript:

MAIAC uses 8 different regional background aerosol models tuned to the AERONET (Aerosol Robotic Network, (Holben et al., 1998)) climatology. Each geographical location has one predefined aerosol model. Aerosol model number 1 is used for Moscow region.

**3.2 Temporal AOT changes in Moscow according to ground-based and satellite data**

We studied temporal AOT changes using MAIAC AOT retrievals and AERONET long-term measurements collocated in time over Moscow_MSU_MO site during a warm May-September period. Fig. 4a shows the time series of AOT at 0.55 µm built for all available Moscow_MSU_MO AERONET and MAIAC data. One can see a satisfactory agreement between the satellite and ground-based observations with the exception of 2002 and 2010 years. The highest AOT were observed in 2010 and 2002 years due to the effects of smoke aerosols from peat and forest fires in Moscow region (Chubarova et al, 2011b). In 2016 the smoke aerosol advection was also observed from the Siberia area (Sitnov et al., 2017) providing an intermediate AOT maximum. Fig.4b shows year-to-year variability of AOT at 0.55 µm only for matching within 1 hour Moscow_MSU_MO AERONET and MAIAC data, and for the cases, when MAIAC regional background aerosol model has been applied. One can see a better agreement between MAIAC AOT and corresponding AERONET AOT data in year-to-year variations. There is a clearly seen decrease in AOT during the last years according to both the MAIAC (when regional model was used) and the AERONET data. The yearly means difference between AERONET and MAIAC data (AOT MAIAC – AOT AERONET) is -0.03 for the all matching data (blue and red lines in Fig 4b) and -0.05 for the matching data with MAIAC regional aerosol model estimates (blue and orange lines in Fig 4b). Fig.4c presents the AOT variations only for the cases of the MAIAC smoke detection. It is seen that the AOT MAIAC overestimation is taken place only for the cases with high AOT>1.

Thus, MAIAC AOT reproduces the absolute AOT values and the long-term AOT decrease in Moscow for the regional background aerosol model while in case of smoke aerosol detection there is a significant overestimation of the annual AOT mean. Therefore, for the further analysis of urban aerosol pollution, we used only the AOT MAIAC retrievals with its attribution to the regional background model for removing large smoke aerosol effects, which are also characterized by significant spatial inhomogeneity.

[Figure]

**Figure 4. The year-to-year variations of AOT at 0.55 μm (May-September, mean values) according to AERONET (Moscow_MSU_MO) and MAIAC data: a) all available AERONET and MAIAC data, b) matching AERONET and MAIAC data for all cases and for regional aerosol model only, c) AOT MAIAC in cases of smoke detection and matching AERONET data.**

*5. It makes sense to work on the style of the presentation. For example, within one paragraph the authors write "One can see: : :: : :" (line numbers 327, 330), "We can see: : :: : :" (line number 333), etc.*

We corrected the style of the presentation: use only one phrase "One can see", and tried to make the changes in other places of the manuscript.

*6. The reference Sever, L., Alpert, P., Lyapustin, A., Wang, Y. and Chudnovsky, A.: An example of aerosol pattern variability over bright surface using high resolution MODIS MAIAC: The eastern and western areas of the Dead Sea and environs, Atmospheric Environment, 165, 359–369, doi:10.1016/j.atmosenv.2017.06.047, 2017 is repeated twice.*

Thank you. We deleted the repeated reference.